# Learning Black-Box Attackers
# with Transferable Priors and Query Feedback

**Jiancheng Yang[1,2]\*, Yangzhou Jiang[1,2,\*], Xiaoyang Huang[1,2], Bingbing Ni[1,2][†], Chenglong Zhao[1,2]**
[1] Shanghai Jiao Tong University, Shanghai 200240, China
[2] MoE Key Lab of Artificial Intelligence, AI Institute, Shanghai Jiao Tong University
{jekyll4168, jiangyangzhou, huangxiaoyang, nibingbing, cl-zhao}@sjtu.edu.cn

## Abstract

This paper addresses the challenging black-box adversarial attack problem, where only classification confidence of a victim model is available. Inspired by consistency of visual saliency between different vision models, a surrogate model is expected to improve the attack performance via transferability. By combining transferability-based and query-based black-box attack, we propose a surprisingly simple baseline approach (named SimBA++) using the surrogate model, which significantly outperforms several *state-of-the-art* methods. Moreover, to efficiently utilize the query feedback, we update the surrogate model in a novel learning scheme, named High-Order Gradient Approximation (HOGA). By constructing a high-order gradient computation graph, we update the surrogate model to approximate the victim model in both forward and backward pass. The SimBA++ and HOGA result in **Le**arnable **B**lack-Box **A**ttack (LeBA), which surpasses previous *state of the art* by considerable margins: the proposed LeBA significantly reduces queries, while keeping higher attack success rates close to 100% in extensive ImageNet experiments, including attacking vision benchmarks and defensive models. Code is open source at `https://github.com/TrustworthyDL/LeBA`.

## 1 Introduction

Deep learning has achieved remarkable success in numerous areas [27, 45, 43, 44]. However, modern deep learning technology is proven vulnerable to adversarial examples for various data modalities [37, 16, 42], which are visually similar examples (compared with natural samples) crafted to fool the deep models. It is important to ensure security of artificial intelligence system, as its increasing employment in high-stakes areas, e,g, autonomous driving [15, 6], healthcare [30, 33] and criminal justice [40, 9], in both cloud and edge computing infrastructure. Research efforts have been made on the adversarial examples, in order to evaluate model robustness [32], improve the security [16], and contribute to model interpretability [12].

Adversarial attack could be categorized into white-box and black-box attack. It appears that vision benchmark models Szegedy et al. [38], He et al. [22], Simonyan & Zisserman [36] are susceptible to white-box attacks [3], even equipped with defense strategies [18]. On the other hand, it is still challenging to attack a victim model in a black-box setting, where only the classification confidence of the victim model is accessible. Prior arts either produce transferable gradients from a white-box surrogate model (*i.e.*, transferability-based approaches [13, 41, 14]), or estimate gradients based on query feedback (*i.e.*, query-based approaches [19, 24]), while these methods still suffer from attack failure or query inefficiency. Even for *state-of-the-art* black-box attack method [10], it needs

[†]Corresponding author: Bingbing Ni.

thousands of queries and suffers from failure cases, especially when attacking strong victim models with defense strategies. In this paper, we aim at an improved black-box attack. As reported in Sec. 4, we significantly improve the query efficiency while keeping higher success rates close to 100%. The performance boosting comes from 1) leveraging transferability-based and query-based approaches, and 2) a learnable surrogate model.

In early research of query-based black-box attack [8, 23, 19], no transferability between vision models is considered. However, as illustrated in Supplementary Figure A1, it is unreasonable to ignore the high consistency between the vision models. By introducing surrogate models as transferable priors, previous methods [10, 20] could improve the attack performance via prior-guided gradient estimation. Although effective, these approaches with fixed surrogate models do not leverage the advances in transferability-based attack [13, 41, 14]. In this study, by combining both transferability-based attack (TIMI [14]) and query-based attack (SimBA [19]), we propose a surprisingly simple yet powerful baseline approach, named SimBA++. It not only improves the query efficiency and attack success rate over TIMI and SimBA, but also outperforms several *state-of-the-art* approaches [24, 20, 10].

However, the query feedback is not well utilized in prior arts. It is expected that the query results divulge information from the victim model, which could potentially improve the attack performance as more queries are available. We update the surrogate model to approximate the victim model with the query feedback. Few prior research suggests a query-efficient update strategy in this scenario. It is possible to train another CNN to refine the gradient from surrogate model; whereas, we assume that it is easier to directly train the back-propagated gradient of surrogate model to be aligned with that of victim model, with the help of high-order gradient computation graph. The proposed update strategy is named High-Order Gradient Approximation (HOGA). As demonstrated in Sec. 4 and Sec. 5, HOGA not only leads to fewer queries, but also learns a transferable surrogate model on new attack data.

Based on the above observation, combining the strong baseline approach SimBA++ with High-Order Gradient Approximation, we propose **Le**arnable **B**lack-Box **A**ttack (LeBA). Extensive experiments on ImageNet dataset [11] validate the superiority of LeBA over previous *state of the art* and our baseline approaches, in terms of query efficiency and attack success rate, especially when attacking strong victim models with defense strategies. Our study highlights the importance of leveraging the query-based and transferability-based black-box attack, as well as the feasibility of learning surrogate models within limited queries.

## 2 Preliminaries

### 2.1 Adversarial Examples and Adversarial Attacks

Adversarial examples are crafted from the natural benign samples with imperceptible changes. We focus on untargeted attack setting, whereas the formulation could be easily extended to targeted attack setting. For a victim image classifier $\mathcal{V}$, given input image $X$, the goal of adversarial attack is to generate an adversarial example $X_{adv}$ to mislead $\mathcal{V}$. It could be formalized into

$$\arg\max(\mathcal{V}(X)) = y, \quad \arg\max(\mathcal{V}(X_{adv})) \neq y,$$
$$s.t. \|X_{adv} - X\|_p \leq \zeta, \tag{1}$$

where $y$ denotes the true label for $X$, and $\|\cdot\|_p$ denotes the $l_p$ norm, and $\zeta$ is the max restriction (or attack budget). According to the exposure degree of victim models, adversarial attack could be categorized into white-box attack and black-box attack.

In white-box setting, adversary has full access to the victim models, including model gradients. Fast Gradient Sign Method (FGSM) [16], as an easy-to-implement white-box attack approach, performs one-step gradient ascent / descent upon input images and restricts the perturbation within an $\zeta$-bounded $l_\infty$ ball. As for the extension of $l_2$ norm, Fast Gradient Method (FGM) [25] generates adversarial example restricted by $l_2$-norm bound of $\zeta$. FGSM and FGM are named one-step methods. Their iterative versions are easily extended, where the adversarial examples are updated in a small step $\epsilon$ in each iteration. Besides, optimization-based methods also be applied to white-box attack, *e.g.*, L-BGFS [37] and C&W [5] attack. In black-box attack, adversary has limited access to victim models, by which complete or partial information is obtained. Prior studies follows either **transferability-based** black-box attack [37, 31, 26] or **query-based** black-box attack [8, 23, 24].

**Threat Model.** In this study, we focus on the black-box attack. Specifically, it is assumed that adversary has surrogate white-box models [10] and query access to classification scores from victim models. We aim at minimizing the average query number of black-box attack under a certain attack budget ($\zeta$). Here, $l_2$ norm is considered as distance metric. The proposed idea of combining both transferability-based and query-based black-box attack is expected to be generally effective; our current method on $l_2$ norm could be extended to $l_\infty$ norm in further studies.

## 2.2 Transferability-Based Black-Box Attacks

Transferability-based black-box attacks are based on the transferability of adversarial examples [37, 31, 26]: Adversarial examples generated by a vision model could potentially fool another vision models. Early research [37] found L-BGFS attack can be transferred between different models on MNIST, and more analysis [31] delves into the transferability on ImageNet [11]. In the transferability-based black-box attack, adversarial examples to a victim model are generated by a white-box surrogate model. Research efforts [13, 41, 14] have been paid in improving the transferability of adversarial examples. Momentum techniques [13] have been introduced in adversarial attack to maintain the stability of gradient. Combining with Iterative FGM, the update formula is

$$\boldsymbol{g}^{t+1} = \mu \cdot \boldsymbol{g}^t + \frac{J\left(X_{adv}^t, y\right)}{\left\|\nabla_X J\left(X_{adv}^t, y\right)\right\|_2},$$
$$X_{adv}^0 = X, \ \boldsymbol{g}^0 = 0, \ X_{adv}^{t+1} = X_{adv}^t + \epsilon \cdot \boldsymbol{g}^{t+1}, \tag{2}$$

where $\mu$ denotes the momentum, and $\epsilon$ denotes the attack step (or "learning rate") of each iteration. This simple technique, momentum iterative (MI) method, is proven effective in practice. Further research [14] argues that using set of translated images to optimize an adversarial example could significantly improve the transferability. This method, named translation-invariant (TI) attack, could be simplified by operating the model gradients $\boldsymbol{W} * \nabla_{\boldsymbol{x}} J(\boldsymbol{x}, y)|_{\boldsymbol{x}=X_{adv}^t}$, where $\boldsymbol{W}$ is a smoothing kernel, *e.g.*, Gaussian kernel, and $*$ denotes convolution. The study [14] suggests that the MI and TI method could be combined into TIMI, which is an important component in our algorithm. TIMI is one of the *state-of-the-art* transferability-based black box attack; however, as shown in Sec. 4, TIMI still suffers from low attack success rates when utilized alone.

## 2.3 Query-Based Black-Box Attacks

Query-based black-box attacks could be divided into score-accessible and decision-accessible settings. We focus on the score-accessible black-box setting, where the classification score of the victim model is available. The decision-accessible setting [4, 7] is beyond the scope of this study, where only discrete classification decision is available.

Given a natural sample $X$, the goal of query-based black-box adversarial attack is to generate an adversarial example $X_{adv}$ to fool a victim model $\mathcal{V}$, with as fewer queries as possible, under $\|X_{adv} - X\|_2 \leq \zeta$. As the gradient of victim models is inaccessible, we argue that there are three key issues for the design of query-based black-box attack, *i.e.*, 1) how to generate a query perturbation in high-dimension space, 2) how to estimate the gradient with less bias and variance, and 3) how to update the query sample with the estimated gradients.

ZOO [8] proposes to use symmetric difference to estimate gradient for every pixel. NES [23] utilizes natural evolution strategy to estimate gradients. Following NES, Bandits attack [24] introduces data-dependent prior and time-dependent prior to improve query efficiency. Instead of the coordinate-wise gradient estimation in ZOO, Bandits attack introduces spherical gradient estimation [21] to integrate the "priors" in the Bandits framework. NATTACK [28] also introduces a gradient estimation framework to improve the attack success over defensive models. RGF [10] follows the same gradient estimation framework.

Different from the previous approaches, SimBA [19] does not estimate gradient explicitly, but adapt a greedy strategy to update the query samples, *i.e.*, randomly samples a query perturbation from a predefined orthonormal basis, either add or subtract it to the target image, and update if the query perturbation decrease / increase the attack loss $J$ (C&W [5] in our study). This simple algorithm surprisingly outperforms several more complex query-based attack, *e.g.*, Bandits [24]. In this study, we use an improved SimBA as a key component in our algorithm. A recent study Square Attack [2] introduces highly-efficient greedy random search for black-box adversarial attack, which could be

regarded as an improved query-based attack over SimBA [19]. Please note that the contributions of Square Attack [2] and our study are orthogonal; They may be integrated together in future studies.

Furthermore, there are also studies exploring to improve query-based attacks with transferability. P-RGF [10] improves query efficiency of RGF with a transfer-based prior. Subspace attack [20] propose to reduce search space of random vectors with gradients from a set of reference models. However, these methods do not explicitly integrate the advances in transferability-based black-box attack (*e.g.*, TIMI); besides, the surrogate model is not updated using the query feedback. Both techniques are proven remarkably effective in our experiments.

## 3 Methodology

### 3.1 Strong Baselines: SimBA+ and SimBA++

We start from improving the SimBA algorithm [19]. The original SimBA samples coordinate-wise perturbation uniformly. However, as the gradient visualization in Supplementary Figure A1, the model contributions from coordinates are absolutely not equal. Considering the consistency of visual saliency between vision models, we introduce a surrogate model $\mathcal{S}$ to guide the query coordinate selection. Instead of sampling uniformly from all coordinates, the sampling probability is proportional to $|M|$, where $M$ is the gradient map from $\mathcal{S}$ via back-propagation, and $|\cdot|$ denotes absolute value. Besides, it is assumed that the spatial neighbors of a pixel coordinate contribute similarly to the model, we introduce the spatial data prior [24], which is implemented with a Gaussian smoothing. Integrating the model-guided query coordinate selection and spatial data prior, each query perturbation $\delta$ is generated by sampling a one-hot coordinate vector $q$ proportional to $|M|$, and then applying $\delta = q * W$, where $W$ denotes a Gaussian convolution kernel. This approach, named *SimBA+*, is regarded as one of the improved SimBA baselines. The complete algorithm is detailed in Supplementary Algorithm A1. Though well-performing compared with prior arts, it still suffers from unsatisfying query efficiency and attack failure, possibly due to cold start and local minima.

We introduce transferability-based attack to solve these issues. Specifically, we apply the translation-invariant attack [14] enhanced by momentum [13], *i.e.*, TIMI, which is powerful and easy to implement. Although it still suffers from attack failure, it provides a warm start for the query-based attack. Besides, by alternatively applying transferability-based attack $\mathcal{T}$ (*i.e.*, TIMI) and query-based attack (*i.e.*, SimBA+), the optimization of adversarial attack is less prone to get stuck in local minima, which suggests the cause of low attack success rates when applying either query-based attack or transferability-based attack alone. This improved SimBA approach, named *SimBA++*, significantly improves the query efficiency and attack success rates, which outperforms SimBA+ and previous *state of the art* [24, 10]. The complete SimBA++ is illustrated in Supplementary Algorithm A2.

Note that we use a clip equation [10] to ensure that $X_{adv}$ does not exceed the attack budget $\zeta$,

$$clip(X_{adv}, X) = \begin{cases} (X + \frac{l_2(X_{adv}, X)}{\zeta} \cdot (X_{adv} - X)), \\ \quad \quad if\, l_2(X_{adv}, X) \geq \zeta; \\ X_{adv}, \; otherwise. \end{cases} \quad (3)$$

We use $\zeta = 16.37 \approx \sqrt{0.001 \times 299 \times 299 \times 3}$ following previous *state of the art* [10].

### 3.2 Learnable Black-Box Attack

The performance of transferability-based attack is highly dependent on the similarity between the surrogate model $\mathcal{S}$ and victim model $\mathcal{V}$. Theoretically, the query feedback from the victim model divulge information from the victim model. This observation leads to **Le**arnable **B**lack-Box **A**ttack (LeBA), where we enhance SimBA++ via updating the surrogate model to approximate the victim model. The complete LeBA algorithm is provided in Algorithm 1. We recommend to use the algorithm to understand the following part. Note that the LeBA (*test*) is exactly the same as SimBA++ if same surrogate models are used.

In order to update the surrogate model within limited query feedback, an efficient learning scheme is needed. To our knowledge, few prior study investigate the learning strategy in this scenario. To this end, we develop High-Order Gradient Approximation (HOGA) to update the surrogate model to simulate both forward and backward pass of victim model. The complete HOGA algorithm

---

**Algorithm 1** **Le**arnable **B**lack-Box **A**ttack (LeBA)

---
**Input:** input image $X$, victim model $\mathcal{V}$, surrogate model $\mathcal{S}$, transferability-based attack $\mathcal{T}$, attack step $\epsilon$, query iteration $n_Q$, buffer $\mathbb{B}$, buffer size $b$, *training / test* mode.
**Output:** (adversarial) example $X_{adv}$.
Initialize $X_{adv} = X; \mathbb{B} = \emptyset$;
Query $\mathcal{V}$ to initialize target probability $P_T$ and loss $J$;
**for** $i$ **in** $\{0, 1, 2, ...\}$ **do**
  **if** $i \bmod n_Q$ **then**
    *{Run transferability-based attack}*
    Run $X'_{adv} = clip(\mathcal{T}(X_{adv}), X)$ {Eq. 3} with $\mathcal{S}$;
    Cache the gradient map $M$ from $\mathcal{S}$;
    Query $\mathcal{V}$ for target probability $P'_T$ and loss $J'$;
  **else**
    *{Run query-based attack}*
    Generate perturbation $\delta$ with Gaussian-smoothed coordinate $q$ (sampled proportional to $|M|$);
    **for** $\alpha$ **in** $\{+\epsilon, -\epsilon\}$ **do**
      $X'_{adv} = clip(X_{adv} + \alpha \cdot \delta, X)$ {Eq. 3};
      Query $\mathcal{V}$ for target probability $P'_T$ and loss $J'$;
      $\mathbb{B}.add(X'_{adv}, X_{adv}, P'_T, P_T)$;
      **if** $J' < J$ **then**
        **break**;
      **end if**
    **end for**
  **end if**
  **if** *training* mode **and** $B.size = b$ **then**
    *{Train the surrogate model $\mathcal{S}$ with HOGA (Supplementary Algorithm A3)}*
    $\mathbb{B} = \emptyset$;
  **end if**
  **if** $J' < J$ **then**
    Update $X_{adv} = X'_{adv}; P_T = P'_T; J = J'$;
  **end if**
  **if** *exceed max query budget* **or** *success* **then**
    **break**;
  **end if**
**end for**
**return** $X_{adv}$.

---

is detailed in Supplementary Algorithm A3. Mathematically, given surrogate model $\mathcal{S}$ and tuple $(\mathbf{X}'_{adv}, \mathbf{X}_{adv}, \mathbf{P}'_T, \mathbf{P}_T)$, where $X_{adv}, P_T$ are the pre-perturbed sample and its target probability from the victim model $\mathcal{V}$, and $X'_{adv}, P'_T$ are for the post-perturbed sample. $\Delta = \mathbf{X}'_{adv} - \mathbf{X}_{adv}$ denotes the query perturbation. As $\Delta$ is a sparse Gaussian-smoothed coordinate-wise perturbation, the gradient of victim model $\mathbf{g}_v$ could be approximated by

$$\mathbf{g}_v(\mathbf{X}'_{adv} - \mathbf{X}_{adv}) = log\mathbf{P}'_T - log\mathbf{P}_T, \tag{4}$$

via first-order Taylor approximation.

Meanwhile, we back-propagate $log\mathbf{S}_T$ to obtain the surrogate gradient $\mathbf{g}_s$ of $X_{adv}$, where $\mathbf{S}_T$ denotes the target probability from the surrogate model $\mathcal{S}$. A perfect surrogate model oughts to hold the property $\mathbf{g}_s \approx \gamma \mathbf{g}_v$, where $\gamma$, named *Gradient Compensation* factor, is a numerical value to compensate the scale difference in $\mathbf{g}_v$ and $\mathbf{g}_s$. We dynamically estimate the value of $\gamma$ with statistics of query results,

$$\gamma \approx \frac{\sum |\mathbf{g}_s(\mathbf{X}'_{adv} - \mathbf{X}_{adv})|}{\sum |\mathbf{g}_v(\mathbf{X}'_{adv} - \mathbf{X}_{adv})|} = \frac{\sum |\mathbf{g}_s(\mathbf{X}'_{adv} - \mathbf{X}_{adv})|}{\sum |log\mathbf{P}'_T - log\mathbf{P}_T|}. \tag{5}$$

Based on this observation, we design the Backward Loss (BL),

$$l_B = MSE(\mathbf{g}_s(\mathbf{X}'_{adv} - \mathbf{X}_{adv}), \gamma(log\mathbf{P}'_T - log\mathbf{P}_T)), \tag{6}$$

which is easy to implement with help of high-order gradient computation built in several modern deep learning framework, *e.g.*, PyTorch [34] and TensorFlow [1]. The Gradient Compensation factor $\gamma$ is demonstrated to reduce queries. In practice, it is beneficial to adaptively update $\gamma$ with momentum (details in Supplementary Algorithm A3).

Furthermore, we observe that it is also beneficial to approximate the forward pass of victim model, which results in Forward Loss (FL),

$$l_F = MSE(\mathbf{S}_T, \mathbf{P}_T). \tag{7}$$

Both FL and BL are proven effective in our experiments, whereas the BL contributes more in training the surrogate model. The proposed High-Order Gradient Approximation (HOGA) uses $\lambda$-weighted Forward Loss and Backward Loss. In Sec. 4 and Sec. 5, we demonstrate the superiority of the proposed LeBA, and analyze the contribution of each component.

# 4 Experiments

## 4.1 Evaluation on ImageNet

**Experiment Setting.** We experiment on ImageNet [11] to demonstrate the efficiency of our algorithm. Due to the fact that the attack difficulty varies for different image subsets, we use the same 1,000 attack images as previous *state of the art* [10] that are selected from ImageNet. We name this dataset as S1. Besides, we randomly select 1,000 images from ImageNet for further validation, which are correctly classified by Inception-V3 [38]. We name this dataset as S2. All of the images are resized to $299 \times 299$. We implement the algorithm with PyTorch [34]. If not specified, the surrogate model used for LeBA is a ResNet-152 [22] pretrained on ImageNet converted from Tensorflow-Slim [17], to follow the setting [10], and the results are reported on S1.

We constrain the max $l_2$ norm of perturbation to $\zeta = \sqrt{0.001 \times 3 \times 299 \times 299} \approx 16.37$ with pixel values in $[0, 1]$, and set the max query budget to 10,000 times. We report the attack success rate (ASR) and average query numbers (AVG.Q). To alleviate the impact of randomness, we report the AVG.Q and ASR over 3 independently repeated experiments, and the complete results are presented in the Supplementary Tables. If the attacker can not successfully fool the victim model within the budget, we consider it a failure case. Following previous studies [10], we report AVG.Q excluding the failure cases. However, in practical scenario, we never know whether a sample could be attacked successfully beforehand, it is thus unreasonable to abandon failure samples when counting query numbers. Thereby, we also report the AVG.Q' including failures in the Supplementary Tables, where failure query numbers are considered as 10,000.

As for hyper-parameters, if not specified, we set the attack step $\epsilon$ to $0.1$, query iteration $n_Q$ to 20, buffer size $b$ to 24, $\lambda = 0.01$ and initial $\gamma = 3.0$ with momentum update. For TIMI, we set iteration numbers $n_T = 10$. Note that $n_Q$ and $n_T$ are tunable hyper-parameters with potential even better performance. Results with various $n_Q$ and $n_T$ are provided in supplementary material.

We compare our methods (SimBA+, SimBA++ and LeBA) with *state-of-the-art* black-box attack methods: transferability-based TIMI [14], query-based NES [23], Bandits$_{TD}$ [24], SimBA [19], Subspace attack [20], and P-RGF series [10]. All the experiments are implemented with the official codes under the same setting. Note that Subspace and P-RGF use surrogate models, which are same as ours. Subspace attack method does not provide $l_2$ norm configuration, thereby we modify the configuration according to that in Bandits. Besides, the SimBA [19] reported is the one with our spatial data prior to compare fairly with our methods, which is better than original SimBA. For TIMI, we keep running iterations of TIMI until the perturbation amount reaches the attack budget $\zeta$.

**Performance Analysis.** Five pretrained vanilla benchmark vision models converted from Tensorflow-Slim [17], including Inception-V3 [38], ResNet-50 [22], VGG-16 [36], Inception-V4 and Inception-ResNet-V2 [39], are regarded as victim models. We report average number of queries (AVG.Q) and attack success rates (ASR) on S1 in Table 1. Compared to previous *state-of-the-art* black-box attack methods, our methods greatly reduces query numbers and increases attack success

Table 1: **Attack performance on ImageNet.** Average number of queries (AVG.Q) and attack success rate (ASR) of the proposed methods and previous *state-of-the-art* black-box attack methods on ImageNet [11], against victim models including Inception-V3 [38], ResNet-50 [22], VGG-16 [36], Inception-V4 [39] and Inception-ResNet-V2 (IncRes-V2) [39]. All the performance is reported using the official codes, under $l_2$ norm and a maximum query number of 10,000. The experiment setting and images are same as previous *state-of-the-art* [10].

| Methods | Inception-V3 | | ResNet-50 | | VGG-16 | | Inception-V4 | | IncRes-V2 | |
|---|---|---|---|---|---|---|---|---|---|---|
| | ASR | AVG.Q | ASR | AVG.Q | ASR | AVG.Q | ASR | AVG.Q | ASR | AVG.Q |
| NES [23] ICML'18 | 88.2% | 1726.3 | 82.7% | 1632.4 | 84.8% | 1119.6 | 80.7% | 2254.3 | 52.5% | 3333.3 |
| Bandits$_{TD}$ [24] ICLR'19 | 97.7% | 836.1 | 93.0% | 765.3 | 91.1% | 275.9 | 96.2% | 1170.9 | 89.7% | 1569.3 |
| Subspace [20] NeurIPS'19 | 96.6% | 1635.8 | 94.4% | 1078.7 | 96.2% | 1085.8 | 94.7% | 1838.2 | 91.2% | 1780.6 |
| RGF [10] NeurIPS'19 | 97.7% | 1313.5 | 97.5% | 1340.2 | 99.7% | 823.2 | 93.2% | 1860.1 | 85.6% | 2135.3 |
| P-RGF [10] NeurIPS'19 | 97.6% | 750.8 | 98.7% | 229.6 | 99.9% | 685.5 | 96.5% | 1095.6 | 88.9% | 1380.2 |
| P-RGF$_D$ [10] NeurIPS'19 | 99.0% | 637.4 | 99.3% | 270.5 | 99.8% | 393.1 | 98.3% | 913.6 | 93.6% | 1364.5 |
| Square [2] ECCV'20 | **99.4%** | 351.9 | 99.8% | 401.4 | **100.0%** | 142.3 | 98.3% | 475.6 | 94.9% | 670.3 |
| TIMI [14] CVPR'19 | 49.0% | - | 68.6% | - | 51.3% | - | 44.3% | - | 44.5% | - |
| SimBA [19] ICML'19 | 97.8% | 874.5 | 99.6% | 873.9 | **100.0%** | 423.3 | 96.2% | 1149.8 | 92.0% | 1516.1 |
| SimBA+ (Ours) | 98.2% | 725.2 | 99.7% | 717.0 | **100.0%** | 365.9 | 96.8% | 946.2 | 92.5% | 1234.7 |
| SimBA++ (Ours) | 99.2% | 295.7 | **99.9%** | 187.3 | 99.9% | 166.0 | 98.3% | 420.2 | 95.8% | 555.1 |
| LeBA (Ours) | **99.4%** | **243.8** | **99.9%** | **178.7** | 99.9% | 145.5 | **98.7%** | 347.4 | **96.6%** | 514.2 |

Table 2: **On the usefulness of learning a surrogate model.** S1 and S2 are two subsets with 1,000 images from ImageNet [11]. We first run the LeBA (*training*) on S1 (as Table 1) and keep the learned surrogate model weight. We then compare the attack performance on both S1 and S2, for SimBA++ and LeBA (*test*). Note that the only difference for these two methods is the surrogate model weight: ImageNet weight for SimBA++, and S1-learned weight for LeBA (*test*).

| Data | Methods | Inception-V3 | | ResNet-50 | | VGG-16 | | Inception-V4 | | IncRes-V2 | |
|---|---|---|---|---|---|---|---|---|---|---|---|
| | | ASR | AVG.Q | ASR | AVG.Q | ASR | AVG.Q | ASR | AVG.Q | ASR | AVG.Q |
| S1 | SimBA++ | 99.2% | 295.7 | **99.9%** | 187.3 | **99.9%** | 166.0 | 98.3% | 420.2 | 95.8% | 555.1 |
| | LeBA (*training*) | **99.4%** | 243.8 | **99.9%** | 178.7 | **99.9%** | 145.5 | **98.7%** | 347.4 | **96.6%** | 514.2 |
| | LeBA (*test*) | **99.4%** | **230.6** | **99.9%** | **172.3** | **99.9%** | **138.5** | 98.4% | **322.4** | **96.6%** | **510.2** |
| S2 | SimBA++ | 99.7% | 183.0 | **100.0%** | 110.4 | **100.0%** | 98.6 | 98.8% | 245.1 | 97.6% | 325.8 |
| | LeBA (*test*) | **99.8%** | **151.3** | **100.0%** | **97.2** | **100.0%** | **96.2** | **98.9%** | **215.9** | 97.6% | **290.8** |

rate. LeBA significantly reduces the average queries and achieves higher success rates close to 100% compared to prior methods, which validates the superiority of our method. Owing to the proposed HOGA learning scheme, our method LeBA further boosts the performance over SimBA++. LeBA improves the performance even more in difficult cases, e.g, Inception-V4 and Inception-ResNet-V2. Besides, improvement from SimBA to SimBA+ demonstrates the effectiveness of using gradient saliency map from surrogate model as guideline for selecting pixel in SimBA. Although TIMI requires no query, it suffers from high attack failure (around 50%). Even for pure query-based methods (SimBA and SimBA+), the attack success is not guaranteed. By alternating the query-based attack and transferability-based attack, our methods (SimBA++ and LeBA) escape the local minima and achieve higher success rates, which are close to 100% in most cases. We also visualize randomly selected LeBA-attacked images in Supplementary Figure A2.

## 4.2 On the Usefulness of Learning a Surrogate Model

**Experiment Setting.** To verify the effectiveness of learning a surrogate model with High-Order Gradient Approximation, we present the results of SimBA++, LeBA (*training* on S1) and LeBA (*test* with fixed surrogate model weight learned on S1) on both dataset S1 and S2 in Table 2.

**Performance Analysis.** Note that the only difference between LeBA (*test*) and SimBA++ here is the weights for surrogate model: ImageNet-pretrained weight for SimBA++, and S1-learned weight for LeBA (*test*). As reported in Table 2, the weights trained on S1 with LeBA is not only 1) **effective**: LeBA (*test*) consistently outperforms LeBA (*training*) and SimBA++, which means the learned weights are better than the fixed initial weights (SimBA++) and dynamically trained weights, but also 2) **transferable**: the learned surrogate model weight on S1 also improves the LeBA (*test*) performance on S2 over SimBA++. The results highlights the life-long learning potential of LeBA: the more attack images, the better attack performance.

Table 3: **The attack performance over the defensive methods**, including JPEG compression [18], guided denoiser [29] and adversarial training [25]. The victim model is Inception-V3 [38].

| Methods | JPEG Compression | | Guided Denoiser | | Adversarial Training | |
|---|---|---|---|---|---|---|
| | ASR | AVG.Q | ASR | AVG.Q | ASR | AVG.Q |
| NES [23] ICML'18 | 14.9% | 2330.9 | 57.6% | 2773.8 | 59.4% | 2773.6 |
| Bandits$_{TD}$ [24] ICLR'19 | 95.8% | 1086.7 | 20.3% | 759.6 | 96.6% | 1121.4 |
| Subspace [20] NeurIPS'19 | 46.7% | 2073.4 | 93.2% | 1619.2 | 93.4% | 1651.7 |
| RGF [10] NeurIPS'19 | 74.4% | 846.9 | 22.0% | 2419.1 | 87.6% | 2095.3 |
| P-RGF$_D$ [10] NeurIPS'19 | 94.8% | 751.2 | 82.6% | 1588.3 | 98.4% | 1092.8 |
| Square [2] ECCV'20 | **98.8%** | 342.3 | 98.2% | 392.6 | 98.5% | 387.6 |
| TIMI [14] CVPR'19 | 48.2% | - | 39.3% | - | 39.2% | - |
| SimBA [19] ICML'19 | 96.0% | 762.8 | 98.0% | 971.6 | 98.0% | 978.0 |
| SimBA+ (Ours) | 96.8% | 663.4 | 98.2% | 797.1 | 98.0% | 779.4 |
| SimBA++ (Ours) | 98.2% | 325.1 | 98.5% | 407.9 | 98.7% | 422.9 |
| LeBA (Ours) | **98.8%** | **273.0** | **98.8%** | **343.6** | **98.9%** | **355.0** |

## 4.3 Attack over Defensive Models

**Experiment Setting.** We further perform attack over defensive models to further demonstrate the query efficiency of our methods, including JPEG Compression [18], Guided Denoiser [29] and Adversarial Training [25]. Note that all results in this section are based on Inception-V3 [38]. Official codes together with the defensive model weights are used.

**Performance Analysis.** The results in Table 3 further demonstrate the superior performance of our method LeBA. Suffering from strong defense strategies, attack success rates decrease greatly for the comparing black-box attack algorithms, but LeBA consistently keeps nearly 99% success rate against the defense strategies in our experiments. Besides, LeBA reduces 64% to 78% queries comparing to previous *state of the art*, P-RGF$_D$. In Supplementary Figure A3, we plot the attack success rate against the number of allowed queries on SimBA [18] and the proposed SimBA+, SimBA++ and LeBA. The figure reveals that SimBA++ and LeBA successfully attacks around half of the images at the beginning of the attack owing to transferability-based attack TIMI. With the help of HOGA learning scheme, LeBA further improves attack efficiency, reducing around 16% queries.

## 5 Ablation Study

In this section, we analyse the function of each component in the proposed LeBA algorithm with several ablation experiments. If not specified, the following ablation experiments are conducted on attack against the Inception-V3 [38] on dataset S1 and keep the same setting as in Sec. 4.1.

**Choice of Surrogate Models.** First, we study the impact of using different surrogate models. Except for ResNet-152 [22] in Sec.4, we select ResNet-101, ResNet-50 and VGG-16 [36] as surrogate models for LeBA to attack against Inception-V3. The results are presented in Table 4: all the surrogate models using for LeBA improves the query efficiency and increases the success rates comparing to unlearnable baseline SimBA++. These experimental results support that the proposed High-Order Gradient Approximations (HOGA) is not sensitive to the choice of surrogate models.

Table 4: **Choice of Surrogate Models,** including ResNet-152, ResNet-101, ResNet-50 [22] and VGG-16 [36]. The victim model is Inception-V3 [38].

| Methods | ResNet-152 | | ResNet-101 | | ResNet-50 | | VGG-16 | |
|---|---|---|---|---|---|---|---|---|
| | ASR | AVG.Q | ASR | AVG.Q | ASR | AVG.Q | ASR | AVG.Q |
| SimBA++ | 99.2% | 295.7 | 99.2% | 288.6 | 99.2% | 279.6 | 98.7% | **336.6** |
| LeBA | **99.4%** | **243.8** | **99.4%** | **231.6** | **99.3%** | **236.3** | **99.1%** | 337.8 |

**High-Order Gradient Approximation.** To verify the necessity of using High-Order Gradient Approximation (HOGA) algorithm, we try to use an additonal UNet (*Additional Net*) [35] instead

of HOGA to learn the gradient of victim model. To be specified, we fix the parameters of surrogate model and add an UNet component, which refines the gradient of the surrogate model. We only update the UNet module using the same loss as in LeBA. In order to maintain a stable beginning of reference gradient output, we make UNet to learn the gradient residual. We report the experiment results on Table 5 (a). The results suggests that using an UNet for gradient approximation does not work well comparing to HOGA algorithm, where LeBA only cost 243.8 average queries while using Additional Net cost 286.5 queries. These experimental results confirm the validity of HOGA, and indicate that it is not easy to approximate a gradient with a train-from-scratch forward model.

Table 5: **(a) High-Order Gradient Approximation (HOGA).** Comparing LeBA (HOGA), *No Learning* (SimBA++) and *Additional Net* to learn the gradient residual. (b) **Gradient Compensation.** Comparing LeBA (adaptive $\gamma$), *No GC* and *Fixed* $\gamma$. (c) **Backward Loss (BL) and Forward Loss (FL).** Comparing LeBA (both BL+FL), *BL only* and *FL only*.

| | Full LeBA | (a) HOGA | | (b) Gradient Compensation | | (c) B & F Loss | |
| | | *No Learning* | *Additional Net* | *No GC* ($\gamma = 1$) | *Fixed* $\gamma = 3$ | *BL only* | *FL only* |
|---|---|---|---|---|---|---|---|
| ASR | **99.4%** | 99.2% | 99.2% | 99.1% | 99.2% | 99.4% | 99.4% |
| AVG.Q | **243.8** | 295.7 | 286.5 | 250.4 | 248.3 | 257.8 | 276.8 |

**Gradient Compensation.**    In this paragraph, we study the impact of Gradient Compensation factor $\gamma$ in HOGA (Supplementary Algorithm A3). The Gradient Compensation factor $\gamma$ is designed to compensate the scale gap in the approximate gradient by querying the victim model and the gradient by back-propagating the surrogate model. We compare no Gradient Compensation (with fixed $\gamma = 1$), fixed $\gamma = 3$ and the proposed adaptive strategy (Supplementary Algorithm A3) with an initial value of 3. Results in Table 5 (b) shows that the LeBA achieve considerable results comparing to SimBA++, with or without Gradient Compensation. Even for no Gradient Compensation version ($\gamma = 1$), LeBA makes an average query number of 250.4, decreasing by 45.3 from SimBA++ (295.7 in Table 1). Moreover, the adaptive $\gamma$ for Gradient Compensation further improves the performance of learning gradient approximation.

**Backward Loss and Forward Loss.**    In this paragraph, we study the impact of Backward Loss (BL), Forward Loss (FL) respectively. We show the results of using BL only, FL only and using both BL and FL (BL+FL) in Table 5 (c). It shows that with Backward Loss only, attacker needs average queries 257.8, which is only a little worse than queries of standard LeBA (BL+FL): 243.8. Besides, Forward Loss also help to learn the gradients, which gets 276.8 queries, which is still better than baseline SimBA++. Nevertheless, combining both Forward Loss and Backward Loss leads to the best performance for LeBA.

# 6   Conclusion

In this paper, by combing transferability-based attack and query-based attack with a learnable surrogate model, we propose **Le**arnable **B**lack-Box **A**ttack (LeBA), which significantly outperforms the previous *state-of-the-art* black-box attack methods by considerable margins, in terms of both query efficiency and attack success rates. By alternating transferability-based attack and query-based attack, the proposed approaches are less prone to get stuck in local minima, which results in fewer queries and higher attack success rates of close to 100% against strong vision benchmark models with defense strategies. With the High-Order Gradient Approximation scheme, we update the surrogate model within limited queries. The learned surrogate model is both 1) effective to improve the performance of transferability-based attack, and 2) transferable to new attack data. The proposed LeBA demonstrates its potential of life-long learning against a victim model, and poses new challenges to the adversarial robustness in a black-box setting.

## Broader Impact

In this paper, by combining transferability-based and query-based adversarial attack, we propose a strong black-box adversarial attack named LeBA, which significantly reduces query numbers against strong victim models while keep high success rates close to 100%. Specifically, it significantly

reduces queries compared to previous *state-of-the-art* query-based black-box attack. For instance, it requires average query numbers of only 243.8, 178.7 and 145.5 to attack Inception-V3, ResNet-50 and VGG-16, respectively.

We introduce two ideas to black-box attack: 1) alternating transferability-based and query-based adversarial attack is surprisingly simple yet effective; 2) learning surrogate model with limited query feedback is feasible. We believe these ideas could benefit the further research in adversarial security of deep vision models. On the other hand, this study poses new challenges to the adversarial robustness in a black-box setting. More research effort should be paid to block the query-based adversarial attackers in real world scenarios.

## Acknowledgments and Disclosure of Funding

This work was supported by National Science Foundation of China (U20B200011, 61976137). Authors appreciate the Student Innovation Center of SJTU for providing GPUs.

## Footnotes

\*These authors have contributed equally.

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
