[Supplementary Material]

# Appendix: *Learning Black-Box Attackers with Transferable Priors and Query Feedback*

**Jiancheng Yang[1,2]\*, Yangzhou Jiang[1,2],⋆, Xiaoyang Huang[1,2], Bingbing Ni[1,2]†, Chenglong Zhao[1,2]**
[1] Shanghai Jiao Tong University, Shanghai 200240, China
[2] MoE Key Lab of Artificial Intelligence, AI Institute, Shanghai Jiao Tong University
{jekyll4168, jiangyangzhou, huangxiaoyang, nibingbing, cl-zhao}@sjtu.edu.cn

## A    Algorithms

We illustrate the complete SimBA+, SimBA++, Learnable Black-Box Attack (LeBA) and High-Order Gradient Approximation (HOGA) in Algorithm A1, Algorithm A2 and Algorithm A3, respectively.

## B    Visualization and More Experiment Results

**Gradient Visualization of Visual Saliency Map.**    Surrogate models for black-box attack in vision models are generally available, since the visual saliency from various vision models is expected to be consistent. In Figure A1, we illustrate the gradients from Inception-V3 [15] and ResNet-152 [9].

Figure A1: **The consistency of visual saliency map from vision benchmark models**. Gradient visualization of Inception-V3 [15] and ResNet-152 [9].

**Visualization of LeBA-attacked Images.**    Randomly selected images before and after adversarial attack by LeBA are illustrated in Figure A2.

**Algorithm A1** SimBA+
---
**Input:** input image $X$, victim model $\mathcal{V}$, surrogate model $\mathcal{S}$, attack step $\epsilon$.
**Output:** (adversarial) example $X_{adv}$.
Initialize $X_{adv} = X$;
Query $\mathcal{V}$ to initialize target probability $P_T$ and loss $J$;
Cache the gradient map $M$ from $\mathcal{S}$;
**for** $i$ **in** $\{0, 1, 2, ...\}$ **do**
  Generate perturbation $\delta$ with Gaussian-smoothed coordinate $q$ (sampled proportional to $|M|$);
  **for** $\alpha$ **in** $\{+\epsilon, -\epsilon\}$ **do**
    $X^{'}_{adv} = clip(X_{adv} + \alpha \cdot \delta, X)$ {Eq. 3};
    Query $\mathcal{V}$ for target probability $P^{'}_T$ and loss $J^{'}$;
    **if** $J^{'} < J$ **then**
      Update $X_{adv} = X^{'}_{adv}$; $P_T = P^{'}_T$; $J = J^{'}$;
      **break**;
    **end if**
  **end for**
  **if** *exceed max query budget* **or** *success* **then**
    **break**;
  **end if**
**end for**
**return** $X_{adv}$.
---

---
**Algorithm A2** SimBA++
---
**Input:** input image $X$, victim model $\mathcal{V}$, surrogate model $\mathcal{S}$, transferability-based attack $\mathcal{T}$, attack step $\epsilon$, query iteration $n_Q$.
**Output:** (adversarial) example $X_{adv}$.
Initialize $X_{adv} = X$;
Query $\mathcal{V}$ to initialize target probability $P_T$ and loss $J$;
**for** $i$ **in** $\{0, 1, 2, ...\}$ **do**
  **if** $i \bmod n_Q$ **then**
    *{Run transferability-based attack}*
    Run $X^{'}_{adv} = clip(\mathcal{T}(X_{adv}), X)$ {Eq. 3} with $\mathcal{S}$;
    Cache the gradient map $M$ from $\mathcal{S}$;
    Query $\mathcal{V}$ for target probability $P^{'}_T$ and loss $J^{'}$;
  **else**
    *{Run query-based attack}*
    Generate perturbation $\delta$ with Gaussian-smoothed coordinate $q$ (sampled proportional to $|M|$);
    **for** $\alpha$ **in** $\{+\epsilon, -\epsilon\}$ **do**
      $X^{'}_{adv} = clip(X_{adv} + \alpha \cdot \delta, X)$ {Eq. 3};
      Query $\mathcal{V}$ for target probability $P^{'}_T$ and loss $J^{'}$;
      **if** $J^{'} < J$ **then**
        **break**;
      **end if**
    **end for**
  **end if**
  **if** $J^{'} < J$ **then**
    Update $X_{adv} = X^{'}_{adv}$; $P_T = P^{'}_T$; $J = J^{'}$;
  **end if**
  **if** *exceed max query budget* **or** *success* **then**
    **break**;
  **end if**
**end for**
**return** $X_{adv}$.
---

**Algorithm A3 H**igh-**O**rder **G**radient **A**pproximation (HOGA)

---

**Input:** surrogate model $\mathcal{S}$, buffer $\mathbb{B}$, $\lambda$ and $\gamma$.
**Output:** updated surrogate model $\mathcal{S}$.
Batch $\mathbf{X}'_{adv}, \mathbf{X}_{adv}, \mathbf{P}'_T, \mathbf{P}_T$ from $\mathbb{B}$;
Compute surrogate target probability $\mathbf{S}_T = \mathcal{S}(\mathbf{X}_{adv})$;
Compute Forward Loss $l_F = MSE(\mathbf{S}_T, \mathbf{P}_T)$;
Create gradient graph and compute $\mathbf{g}_s = \frac{\partial log\mathbf{S}_T}{\partial \mathbf{X}_{adv}}$;
Compute Backward Loss $l_B$ using
$l_B = MSE(\mathbf{g}_s(\mathbf{X}'_{adv} - \mathbf{X}_{adv}), \gamma(log\mathbf{P}'_T - log\mathbf{P}_T))$;
Back-propagate $l_B + \lambda l_F$ with high-order gradient;
Update $\gamma = 0.9 \cdot \gamma + 0.1 \cdot \frac{\sum |\mathbf{g}_s(\mathbf{X}'_{adv} - \mathbf{X}_{adv})|}{\sum |log\mathbf{P}'_T - log\mathbf{P}_T|}$;
Optimize $\mathcal{S}$;
**return** $\mathcal{S}$.

---

Table A1: AVG.Q' version of main text Table 1. **Attack performance on ImageNet.** Average number of queries (AVG.Q') and attack success rate (ASR) of the proposed methods and previous *state-of-the-art* black-box attack methods on ImageNet [3], against victim models including Inception-V3 [15], ResNet-50 [9], VGG-16 [14], Inception-V4 [16] and Inception-ResNet-V2 (IncRes-V2) [16]. All the performance is reported using the official codes, under $l_2$ norm and a maximum query number of 10,000. The experiment setting and images are same as previous *state-of-the-art* [2].

| Methods | Inception-V3 | | ResNet-50 | | VGG-16 | | Inception-V4 | | IncRes-V2 | |
| | ASR | AVG.Q' | ASR | AVG.Q' | ASR | AVG.Q' | ASR | AVG.Q' | ASR | AVG.Q' |
| --- | --- | --- | --- | --- | --- | --- | --- | --- | --- | --- |
| NES [10] | 88.2% | 2702.6 | 82.7% | 3080.0 | 84.8% | 2469.4 | 80.7% | 3749.2 | 52.5% | 6500.0 |
| Bandits$_{TD}$ [11] | 97.7% | 1046.9 | 93.0% | 1411.7 | 91.1% | 1141.3 | 96.2% | 1506.4 | 89.7% | 2437.7 |
| Subspace [8] | 96.6% | 1920.2 | 94.4% | 1578.3 | 96.2% | 1424.5 | 94.7% | 2270.8 | 91.2% | 2503.9 |
| RGF [2] | 97.7% | 1513.3 | 97.5% | 1556.7 | 99.7% | 850.7 | 93.2% | 2413.6 | 85.6% | 3267.8 |
| P-RGF [2] | 97.6% | 972.8 | 98.7% | 356.6 | 99.9% | 694.8 | 96.5% | 1407.3 | 88.9% | 2337.0 |
| P-RGF$_D$ [2] | 99.0% | 731.0 | 99.3% | 338.6 | 99.8% | 412.3 | 98.3% | 1068.1 | 93.6% | 1917.2 |
| Square [1] ECCV'20 | **99.4%** | 409.8 | 99.8% | 420.6 | **100.0%** | 142.3 | 98.3% | 637.5 | 94.9% | 1146.1 |
| TIMI [4] | 49.0% | - | 68.6% | - | 51.3% | - | 44.3% | - | 44.5% | - |
| SimBA [7] | 97.8% | 1075.3 | 99.6% | 910.4 | **100.0%** | 423.3 | 96.2% | 1486.1 | 92.0% | 2194.8 |
| SimBA+ (Ours) | 98.2% | 892.1 | 99.7% | 744.8 | **100.0%** | 365.9 | 96.8% | 1235.9 | 92.5% | 1892.1 |
| SimBA++ (Ours) | 99.2% | 373.3 | **99.9%** | 197.1 | 99.9% | 175.8 | 98.3% | 583.1 | 95.8% | 951.8 |
| LeBA (Ours) | **99.4%** | **302.3** | **99.9%** | **188.5** | 99.9% | **155.4** | **98.7%** | **472.9** | **96.6%** | **836.7** |

**Attack Success Rate against Number of Queries.** In Figure A3, we plot the attack success rate against the number of allowed queries on SimBA [6] and the proposed SimBA+, SimBA++ and LeBA. The figure, based on the results on attack over defensive models, reveals that SimBA++ and LeBA successfully attacks around half of the images at the beginning of the attack owing to transferability-based attack TIMI. With the help of HOGA learning scheme, LeBA further improves attack efficiency, reducing around 16% queries.

**Numbers of Queries including Failures.** If the attacker can not successfully fool the victim model within the budget, we consider it a failure case. Following previous studies [2], we report AVG.Q excluding the failure cases in the main text. However, in practical scenario, we never know whether a sample could be attacked successfully beforehand, it is thus unreasonable to abandon failure samples when counting query numbers. Thereby, we also report the AVG.Q' including failures (in Table A1, A2, A3, A4, A5), where failure query numbers are considered as 10,000. Please note that AVG.Q' and AVG.Q can be converted easily:

$$AVG.Q = \frac{AVG.Q' - (1 - ASR) \times 10000}{ASR}. \tag{A1}$$

**Independently Repeated Results of Controlled Experiments.** To alleviate the impact of randomness, we report the AVG.Q' and ASR over 3 independently repeated experiments on ImageNet, with vision benchmark models (main text Table 1) and defensive models (main text Table 3), in Table A6 and Table A7, respectively.

Figure A2: **Randomly selected images before and after adversarial attack by LeBA.** The numbers above figures denote *target probability*, $l_2$ *norm distance from original image*, *attack step*.

**Ablation of $n_Q$ and $n_T$ in TIMI.** We depict the experiment results on tuning attack intervals $n_Q$ and $n_T$ in Table A8. Note that with careful tuning $n_Q$ and $n_T$, it may lead to even better performance. As TIMI is not the main contribution of our study, we have not heavily tuned $n_Q$ and $n_T$.

Figure A3: **Illustration of the query efficiency.** The attack success rate against the defense [6, 13, 5], versus the number of allowed queries on SimBA [7], SimBA+, SimBA++ and LeBA.

Table A2: AVG.Q' version of main text Table 2. **On the usefulness of learning a surrogate model.** S1 and S2 are two subsets with 1,000 images from ImageNet [3]. We first run the LeBA (*training*) on S1 and keep the learned surrogate model weight. We then compare the attack performance on both S1 and S2, for SimBA++ and LeBA (*test*). Note that the only difference for these two methods is the surrogate model weight: ImageNet weight for SimBA++, and S1-learned weight for LeBA (*test*).

| Data | Methods | Inception-V3 | | ResNet-50 | | VGG-16 | | Inception-V4 | | IncRes-V2 | |
| | | ASR | AVG.Q' | ASR | AVG.Q' | ASR | AVG.Q' | ASR | AVG.Q' | ASR | AVG.Q' |
|---|---|---|---|---|---|---|---|---|---|---|---|
| S1 | SimBA++ | 99.2% | 373.3 | **99.9%** | 197.1 | **99.9%** | 175.8 | 98.3% | 583.1 | 95.8% | 951.8 |
| | LeBA (*tra.*) | **99.4%** | 302.3 | **99.9%** | 188.5 | **99.9%** | 155.4 | **98.7%** | **472.9** | **96.6%** | 836.7 |
| | LeBA (*test*) | **99.4%** | **289.2** | **99.9%** | **182.1** | **99.9%** | **148.4** | 98.4% | 477.2 | **96.6%** | **832.9** |
| S2 | SimBA++ | 99.7% | 212.5 | **100.0%** | 110.4 | **100.0%** | 98.6 | 98.8% | 362.2 | **97.6%** | 558.0 |
| | LeBA (*test*) | **99.8%** | **171.0** | **100.0%** | **97.2** | **100.0%** | **96.2** | **98.9%** | **323.5** | **97.6%** | **523.8** |

Table A3: AVG.Q' version of main text Table 3. **The attack performance over the defensive methods**, including JPEG compression [6], guided denoiser [13] and adversarial training [12]. The victim model is Inception-V3 [15].

| Methods | JPEG Compression | | Guided Denoiser | | Adversarial Training | |
| | ASR | AVG.Q' | ASR | AVG.Q' | ASR | AVG.Q' |
|---|---|---|---|---|---|---|
| NES [10] | 14.9% | 8857.3 | 57.6% | 5837.7 | 59.4% | 5707.5 |
| Bandits$_{TD}$ [11] | 95.8% | 1461.1 | 20.3% | 8124.2 | 96.6% | 1423.3 |
| Subspace [8] | 46.7% | 6298.3 | 93.2% | 2189.1 | 93.4% | 2202.7 |
| RGF [2] | 74.4% | 3190.1 | 22.0% | 8332.2 | 87.6% | 3075.5 |
| P-RGF$_D$ [2] | 94.8% | 1232.1 | 82.6% | 3051.9 | 98.4% | 1235.3 |
| Square [1] | **98.8%** | 458.2 | 98.2% | 565.5 | 98.5% | 531.8 |
| TIMI [4] | 48.2% | - | 39.3% | - | 39.2% | - |
| SimBA [7] | 96.0% | 1132.3 | 98.0% | 1152.2 | 98.0% | 1158.4 |
| SimBA+ (Ours) | 96.8% | 962.2 | 98.2% | 962.8 | 98.0% | 963.8 |
| SimBA++ (Ours) | 98.2% | 499.2 | 98.5% | 551.8 | 98.7% | 547.4 |
| LeBA (Ours) | **98.8%** | **389.7** | **98.8%** | **459.5** | **98.9%** | **461.1** |

Table A4: AVG.Q' version of main text Table 4. **Choice of Surrogate Models,** including ResNet-152, ResNet-101, ResNet-50 [9] and VGG-16 [14]. The victim model is Inception-V3 [15].

| Methods | ResNet-152 | | ResNet-101 | | ResNet-50 | | VGG-16 | |
| | ASR | AVG.Q' | ASR | AVG.Q' | ASR | AVG.Q' | ASR | AVG.Q' |
|---|---|---|---|---|---|---|---|---|
| SimBA++ | 99.2% | 373.3 | 99.2% | 366.3 | 99.2% | 357.4 | 98.7% | 462.2 |
| LeBA | **99.4%** | **302.3** | **99.4%** | **290.2** | **99.3%** | **304.6** | **99.1%** | **424.8** |

Table A5: AVG.Q' version of main text Table 5. **(a) High-Order Gradient Approximation (HOGA).** Comparing LeBA (HOGA), *No Learning* (SimBA++) and *Additional Net* to learn the gradient residual. (b) **Gradient Compensation.** Comparing LeBA (adaptive $\gamma$), *No GC* and *Fixed $\gamma$*. (c) **Backward Loss (BL) and Forward Loss (FL).** Comparing LeBA (both BL+FL), *BL only* and *FL only*.

| | Full LeBA | (a) HOGA | | (b) Gradient Compensation | | (c) B & F Loss | |
| | | *No Learning* | *Additional Net* | *No GC ($\gamma = 1$)* | *Fixed $\gamma = 3$* | *BL only* | *FL only* |
|---|---|---|---|---|---|---|---|
| ASR | **99.4%** | 99.2% | 99.2% | 99.1% | 99.2% | 99.4% | 99.4% |
| AVG.Q' | **302.3** | 373.3 | 364.2 | 338.1 | 326.4 | 316.3 | 335.1 |

Table A6: **Independently repeated attack experiments on ImageNet.** 3 experiments of main text Table 1 are reported, in terms of average number of queries (AVG.Q') and attack success rate (ASR) of SimBA [7] and the proposed methods on ImageNet, against victim models including Inception-V3 [15], ResNet-50 [9], VGG-16 [14], Inception-V4 [16] and Inception-ResNet-V2 (IncRes-V2) [16].

| | Inception-V3 | | ResNet-50 | | VGG-16 | | Inception-V4 | | IncRes-V2 | |
| Methods | ASR | AVG.Q' | ASR | AVG.Q' | ASR | AVG.Q' | ASR | AVG.Q' | ASR | AVG.Q' |
|---|---|---|---|---|---|---|---|---|---|---|
| | 97.8% | 1070.1 | 99.4% | 909.8 | 100.0% | 425.0 | 96.2% | 1487.1 | 92.1% | 2165.1 |
| SimBA [7] | 97.6% | 1090.2 | 99.7% | 906.6 | 100.0% | 420.0 | 95.9% | 1492.9 | 92.0% | 2203.6 |
| | 98.0% | 1065.5 | 99.6% | 914.7 | 100.0% | 425.0 | 96.4% | 1478.4 | 92.0% | 2215.6 |
| | 98.2% | 897.7 | 99.7% | 747.6 | 100.0% | 364.6 | 96.9% | 1233.9 | 92.4% | 1908.0 |
| SimBA+ | 98.2% | 892.6 | 99.8% | 745.2 | 100.0% | 366.0 | 96.7% | 1239.8 | 92.8% | 1856.3 |
| | 98.1% | 886.0 | 99.7% | 741.5 | 100.0% | 367.0 | 96.9% | 1234.0 | 92.3% | 1912.0 |
| | 99.1% | 379.2 | 99.9% | 204.3 | 99.9% | 179.0 | 98.2% | 595.5 | 95.9% | 966.3 |
| SimBA++ | 99.2% | 375.7 | 99.9% | 192.6 | 99.9% | 174.8 | 98.4% | 568.5 | 96.1% | 926.5 |
| | 99.2% | 364.9 | 99.9% | 194.4 | 100.0% | 173.4 | 98.2% | 585.3 | 95.3% | 962.5 |
| | 99.4% | 301.2 | 99.8% | 186.5 | 99.9% | 156.7 | 98.9% | 460.7 | 96.3% | 861.4 |
| LeBA | 99.2% | 304.1 | 100.0% | 195.4 | 99.9% | 152.8 | 98.5% | 457.8 | 96.8% | 786.1 |
| | 99.4% | 301.7 | 99.9% | 179.8 | 99.9% | 156.5 | 98.7% | 501.0 | 96.6% | 862.6 |

Table A7: **Independently repeated attack experiments over the defensive methods**, including JPEG compression [6], guided denoiser [13] and adversarial training [12]. 3 experiments of main text Table 3 are reported.

| | JPEG Compression | | Guided Denoiser | | Adversarial Training | |
| Methods | ASR | AVG.Q' | ASR | AVG.Q' | ASR | AVG.Q' |
|---|---|---|---|---|---|---|
| | 96.0% | 1142.8 | 98.0% | 1151.1 | 98.0% | 1165.6 |
| SimBA [7] | 96.2% | 1113.5 | 98.1% | 1149.1 | 97.7% | 1135.4 |
| | 95.9% | 1140.6 | 98.0% | 1156.4 | 97.8% | 1174.2 |
| | 96.8% | 962.2 | 98.3% | 947.1 | 98.0% | 974.9 |
| SimBA+ | 96.9% | 957.7 | 98.0% | 980.1 | 98.2% | 954.7 |
| | 96.8% | 966.6 | 98.2% | 961.3 | 98.2% | 961.7 |
| | 98.2% | 489.3 | 98.6% | 555.4 | 98.5% | 557.3 |
| SimBA++ | 98.4% | 487.3 | 98.5% | 540.5 | 98.8% | 534.3 |
| | 98.0% | 520.8 | 98.5% | 559.7 | 98.7% | 550.8 |
| | 98.5% | 418.9 | 99.0% | 453.4 | 98.7% | 485.2 |
| LeBA | 98.7% | 392.2 | 98.8% | 451.0 | 99.1% | 460.3 |
| | 99.1% | 357.9 | 98.7% | 474.2 | 99.0% | 437.8 |

Table A8: **Tuning attack intervals $n_Q$ and $n_T$.** Attack performance (ASR and AVG.Q') varying attack interval pairs.

| $n_T$ | $n_Q = 10$ | | $n_Q = 20$ | | $n_Q = 30$ | |
|---|---|---|---|---|---|---|
| 5 | 99.2% | 342.1 | 99.1% | 363.1 | 99.2% | 348.5 |
| 10 | 99.4% | 283.8 | 99.4% | 302.3 | 99.3% | 311.4 |
| 15 | 99.4% | 263.2 | 99.5% | 261.7 | 99.4% | 278.9 |

## Footnotes

\*These authors have contributed equally.

†Corresponding author: Bingbing Ni.