[Reviews · NeurIPS 2020]

Review 1

Summary and Contributions: This paper proposes several modifications to the black-box adversarial attack SimBA to improve its query efficiency. The authors find that by using a surrogate model for biased pixel sampling, alternating optimization with a transfer-based attack (TIMI), and updating the surrogate model to more closely match the victim model, query efficiency can be drastically improved against both undefended and defended models in the untargeted attack setting.

Strengths: 1. The empirical evaluation is thorough. The authors attacked several victim models on ImageNet and both the average query reduction and success rate increase are consistent across all victim models. The authors also performed ablation study to demonstrate the effectiveness of each modification to SimBA. 2. Reduction in average query is significant. Compared to both the baseline method SimBA and other SOTA attack algorithms, both SimBA++ and LeBA consistently achieve a close to 50% reduction in average queries with higher success rate. Even against other methods that use a transfer prior (such as Subspace Attack and P-RGF), the improvement is significant and highly non-trivial. 3. The attack remains effective against defended models. Prior work (such as Low Frequency Adversarial Perturbation by Guo et al., UAI 2019 and NAttack by Li et al., ICML 2019) have shown that defended models can hinder the effectiveness of black-box attack algorithms. The proposed method can bypass defenses such as JPEG compression and adversarial training, demonstrating its applicability in realistic scenarios.

Weaknesses: 1. The technical contribution of this paper more closely resembles a "bag of tricks" for improving query efficiency of black-box attacks rather than being scientific. As such, it may be of limited interest to readers outside of the subject area. 2. The paper did not consider extension to targeted attack, which is arguably more meaningful in real world settings. Although I believe the proposed method can remain effective, the lack of empirical evidence is a clear weakness that should be addressed.

Correctness: Based on my understanding, the evaluation methodology of this paper is sound and follows common practice when evaluating black-box attacks.

Clarity: The algorithmic aspect of this paper is mostly deferred to the appendix, which I believe is a significant weakness. I strongly suggest the authors to restructure the paper to make its technical component more self-contained, perhaps by eliminating some redundant details in sections 1 and 2.

Relation to Prior Work: This paper is along a line of recent works that exploit a transfer prior to improve the query efficiency of black-box attacks. The main difference is its superior empirical performance. I feel that this relationship to prior work is not clearly explained and emphasized.

Reproducibility: Yes

Additional Feedback: I strongly encourage the authors to address the issues relating to targeted attack and improving paper organization. Nevertheless, from a performance standpoint, this paper seems very strong to me. ------------------------------------------------------ I read the author response and some of my concerns have been addressed. While other reviewers maintain that the technical contribution of this paper does not warrant publication, I feel that given the superior performance and thorough evaluation, this paper could still serve as a strong baseline for future research.


Review 2

Summary and Contributions: This study develops an attack algorithm against image recognition models in the black-box setting (probability vectors are visible to attackers). The baseline attack algorithm is SimBA presented in ICML19 and additional techniques are newly introduced in this study. One is High-Order Gradient Approximation (HOGA). Given a surrogate model and query responses when adversarial examples are given to the target model, HOGA evaluates the gradient to update the surrogate model. Another technique is Gradient Compensation, which estimates the scale factor of the gradient. This helps to improve the query efficiency of the attack.

Strengths: This paper shows strong experimental results. The attack success rate and query effiecien is significantly improved compared to existing attack methods.

Weaknesses: The technical originality is incremental. The proposed attack scheme basically follows an existing attack method.

Correctness: The empirical methodology seems to be correct.

Clarity: The difference from the existing attack scheme (SimBA) is not large whereas a large space is not spent for explanation of the difference. The clarity could be much improved.

Relation to Prior Work: Relation to prior work is clear.

Reproducibility: Yes

Additional Feedback: This work is somewhat incremental but the experimental results are strong. I think the rational behind the two proposed scheme, HOGA and Gradient compensation, should be clarified more to identify why the proposed scheme works successfully. The ablation study simply explains the results only and do not give good justification of the proposed scheme.


Review 3

Summary and Contributions: The paper presents an approach that combines transfer-based (TIMI) and query-based L2 adversarial attacks (SimBA) that results in a method SimBA++ that outperforms the previous approaches in terms of the success rate and query efficiency. Then the authors suggest a method how to fine-tune the surrogate model using queries obtained from the target model which gives an additional small improvement in terms of the success rate and query efficiency.

Strengths: - The paper presents a more query-efficient way to combine transfer-based and query-based L2 adversarial attacks outperforming the previous approaches (but see the comment below in “Weaknesses” about an additional attack and the significance of this result).

Weaknesses: - The paper’s main idea of mixing transfer-based and query-based attacks is not novel. There have already been multiple papers based on this idea [9, 19]. This paper simply proposes to combine the best transfer-based attack (TIMI) and one of the best L2 query-based attacks (SimBA) which results in SimBA++ that gives the main gain over the previous approaches reported in the paper. If TIMI alone already achieves, e.g., 68.6% success rate on ResNet-50, then counting all these images as requiring only 1 query is of course going to significantly reduce the average number of queries. Yes, this approach leads to a new state of the art, but I don’t think that it gives any interesting new insight. The paper could be a good fit as a workshop paper. - Note that the approach based on fine-tuning the surrogate model (LeBA) gives only marginal improvements over SimBA++ in terms of the number of required queries. Namely, the average reduction is ~14% based on the numbers from Table 1. Even if we assume that it is a significant improvement, the overall algorithm is a bit questionable since it becomes effective only if the attack is performed simultaneously on many inputs (e.g. 1000 inputs). I don’t think it’s a meaningful attack setting as it seems quite restrictive to assume that the attacker needs to misclassify 1000 images simultaneously. Please correct me if I wrongly understood the precise way how the queries for different images are utilized for HOGA. - A comparison to the Square Attack [a] is missing which is a state-of-the-art L2 attack among those that do not rely on surrogate models. For example, see Table 2 therein: the success rate already reaches nearly 100% and the number of queries is on the same order of magnitude, despite that they use the L2 norm of 5, while the current paper uses perturbations of L2 norm of 16.37. In my opinion, this comparison is crucial to include in any case since it would be necessary to show that the proposed method at least outperforms the existing L2 attacks that do not use extra knowledge of surrogate models. [a] Square Attack: a query-efficient black-box adversarial attack via random search, ECCV 2020, https://arxiv.org/abs/1912.00049

Correctness: Yes.

Clarity: Yes, with a few typos and some statements that would need a correction (see below).

Relation to Prior Work: Yes.

Reproducibility: Yes

Additional Feedback: Line 22, Line 93: “Research efforts have been *paid*” -- Incorrect verb in this context. Line 28-30: “Prior arts either produce transferable gradients from a white-box surrogate model (i.e., transferability-based approaches [12, 38, 13]), or estimate gradients based on query feedback (i.e., query-based approaches [18, 23])” -- The approach of [18], SimBA, do not estimate the gradient but rather perform a greedy search. Line 38-39: “However, as illustrated in Supplementary Figure A1, it is unreasonable to ignore the high consistency between the vision models.” -- I think it is unreasonable only if one aims at generating black-box adversarial examples on ImageNet or similar very popular datasets. However, it’s always been just a proof of concept and not an end goal. For a real-world task, surrogate models are scarce. Imagine, you would like to generate black-box adversarial attacks for an ad detection system [b], how would you even obtain a surrogate model for it? One can always argue that it is possible to collect the data from scratch, train some CNN on it and try to transfer the adversarial examples. However, it’s not what is really done in the current line of research on black-box attacks that use surrogate models. Instead, one usually takes a very similar model (all deep CNNs are quite similar to each other) trained on exactly the same dataset (ImageNet) and often with similar hyperparameters. Yes, this setup can be interesting and valuable in research for new insights, but one should keep in mind that this is a huge simplification of the real-world setup and one should ideally adjust the text of the paper accordingly to reflect this. Line 46: “However, the query feedback is not well utilized.” -- This sentence seems to be unfinished. Line 51: “we assume that it is easier to directly train the back-propagated gradient of surrogate model to be aligned” -- It’s clear what is meant, but it would be better to rephrase such that it’s clear that one *trains a model*, and not its gradient. Line 64: “Adversarial examples are crafted from the natural benign samples with *imperceptible* changes.” -- This statement is not true. Adversarial examples need not be imperceptible in general (e.g. see L0 perturbations or adversarial patches). Moreover, the examples shown in the appendix clearly show that the generated by the proposed method adversarial perturbations *are* visible. Line 85: “attack is expected generally effective” -> “attack is expected to be generally effective” Line 90: “attack can be transfer” -> “attack can be transferred” Lines 94-95: “Momentum techniques [12] have been introduced in adversarial attack to maintain the stability of gradient.” -- What is “stability of gradient”? It’s not an established term, does it have any clear definition? If not, I would remove this statement. Line 112-115: “we argue that there are three key issues: … 2) how to estimate the gradient with less bias and variance, and 3) how to update the query sample with the estimated gradients.” -- Again, it’s somehow assumed that one has to necessarily perform gradient estimation for black-box attacks. This is not true, see, e.g., [19] (used heavily in this paper), [c]. Line 147-148: “Though well-performing compared with prior arts, it still suffers from unsatisfying query efficiency and attack failure, possibly due to cold start and local minima.” -- The meaning of “cold start” is unclear to me. What is a hot start? Point close to the global minimum? Line 152-153: “Besides, by alternatively applying transferability-based attack T (i.e., TIMI) and query-based attack (i.e., SimBA+), the optimization of adversarial attack is less prone to get stuck in local minima” -- Is there *any* evidence to claim that the attack really stocks in local minima? If yes, then this evidence should be presented, if not, then this (and all similar phrases throughout the paper) have to be removed. Line 166-167: “We recommend to use the algorithm to understand the following part.” -- “to use” -> “to check”, maybe? And if the algorithm is so important, it would make sense to move it to the main part and instead put some ablations studies (e.g., about the U-Net) to the appendix. Line 243: “4.2 On the Usefulness of Learning” -- The title name should be made more specific (i.e. what kind of learning you mean). Table 4: The surrogate models being tested are not diverse enough. There are 3 ResNets (ResNet-50, ResNet-101 and ResNet-152) and VGG-16. It would be better to evaluate different architectures for surrogate models. Related to this, using VGG-16 instead of ResNet-101 increases the required number of queries by 50%. It would be good to reflect this observation in the text which implies that the choice of the surrogate model has an influence. Appendix, Line 4: “We” shouldn’t be capitalized Appendix, Line 14: it’s not clear which exactly saliency map method was used for this experiment. Figure A1: I would recommend avoiding using photographs of people in the paper without their consent. I understand that these are images from ImageNet, but I am not sure whether it is a good idea to explicitly include these photographs in a paper. ---------------------------- Update after reading the rebuttal ---------------------------- Thanks for providing a detailed rebuttal. A few comments below after reading the rebuttal: - Including the results on targeted attacks, and also a more detailed comparison to the literature (NAttack and Square Attack) has strengthened the experimental evaluation of the paper. - I agree with the authors that just focusing on a single threat model -- L2 in this case -- is totally fine as, say, L2 and Linf perturbation sets have a different structure and it may not be straightforward in general to adapt a method from one norm to another. - However, my main concern still remains. My personal takeaway from this paper is: SimBA (prior work) + TIMI (prior work) lead to a new state of the art. But I don't think it's an interesting result which would be sufficient for a NeurIPS paper. And I still believe that the marginal improvements from SimBA to SimBA+ (implementing an importance sampling scheme for SimBA) and from SimBA++ to LeBA (fine-tuning the surrogate model on the outputs of the target model) are too small to consider them as an interesting new contribution (see the 2 points below for more details). - The improvement from SimBA+ is obtained *before* adding TIMI to the method (i.e. SimBA++) and I suspect that the improvement can be significantly reduced if one first adds TIMI to the method and only then adds the importance sampling scheme. - I still think that the assumptions of LeBA are not very realistic. E.g., the authors in the rebuttal refer to Table 2 to show that LeBA (training) still improves the query efficiency. However, this scheme still requires simultaneously fooling many (e.g., 1000 as in Table 2) images so that the surrogate model can be fine-tuned using the output of these queries. I do not think that simultaneously fooling 1000 images is a "practical scenario". To summarize, I appreciate the efforts of the authors to provide a more thorough experimental evaluation but I stick to my original score 4/10 since my main concerns remain in place.


Review 4

Summary and Contributions: The authors propose a novel black-box attack for deep neural networks. They combine transferability-based and query-based attack, and efficiently update the surrogate model to reduce the queries to the target model. The experiment results on vanilla DNNs as well as defend ones demonstrate that the attack is stronger than other baselines.

Strengths: 1. The paper is well-written and easy to follow. The proposed approach is simple and easy to be reproduced. 2. Although combine the transferability-based and query-based attack is not novel, the authors integrate a more advanced transferability-based attack – TIMI and the improvement is significant. 3. Updating the surrogate model using the query feedback is novel and it can dramatically reduce the number of queries and improve the attack success rate.

Weaknesses: 1. Evaluating on l_2 norm is not sufficient since most of the papers conduct attacking experiments considering l_ \infty as the distance metric and the black-box attack cannot achieve good results on defended models on l_\infty distance. 2. It would be clearer if the authors also list comparison results with some representative white-box attacks like PGD since multiple steps (e.g., 2000 steps) PGD has achieved superior attack success rate on some defended DNNs. 3. Nattack [1] is a strong black-box query-based attack, I would like to see the comparison results with it. 4. Although integrating TIMI into a query-based attack is incremental, I like the idea of update the surrogate model. I would like to raise my rating score if the authors can address my questions during the rebuttal period. [1] Li, Yandong, et al. "Nattack: Learning the distributions of adversarial examples for an improved black-box attack on deep neural networks." arXiv preprint arXiv:1905.00441 (2019).

Correctness: Yes.

Clarity: Yes

Relation to Prior Work: yes.

Reproducibility: Yes

Additional Feedback: After carefully reading the comments and the discussion, I agree with R1 that the empirical performance is a strong point of this paper even though the technical contributions are limited. The authors’ feedback addressed my questions during rebuttal period. I would like to raise my point to 6/10. I like the idea of updating surrogate model to more close to victim model.

[Author Response · NeurIPS 2020]

Table 1: Attack performance of two counterparts following the same setting as main Table 1.

| Methods | Inception-V3 | | ResNet-50 | | VGG-16 | | Inception-V4 | | IncRes-V2 | |
|---|---|---|---|---|---|---|---|---|---|---|
| | ASR | AVG.Q | ASR | AVG.Q | ASR | AVG.Q | ASR | AVG.Q | ASR | AVG.Q |
| NATTACK Li et al. (2019) ICML'19 | 98.2% | 936.1 | 99.5% | 621.3 | 99.7% | 313.0 | 97.5% | 1826.2 | 92.8% | 2688.5 |
| Square Andriushchenko et al. (2020) ECCV'20 | 99.4% | 351.9 | 99.8% | 401.4 | **100.0%** | **142.3** | 98.3% | 475.6 | 94.9% | 670.3 |
| SimBA++ (Ours) | 99.2% | 295.7 | **99.9%** | 187.3 | 99.9% | 166.0 | 98.3% | 420.2 | 95.8% | 555.1 |
| LeBA (Ours) | **99.4%** | **243.8** | **99.9%** | **178.7** | 99.9% | 145.5 | **98.7%** | **347.4** | **96.6%** | **514.2** |

**To all reviewers**: We deeply appreciate all reviewers for the high-quality reviews, especially in the shadow of COVID-
19. All discussions in this rebuttal will be included in the revised paper. All typos and writing issues will be refined.

*Counterpart Methods*: As suggested by reviewers, in Table 1, we additionally report the performance of NATTACK Li
et al. (2019) and Square Attack Andriushchenko et al. (2020) (current SOTA without surrogate models). Although under
different threat models (we use surrogate models while they do not), the proposed SimBA++ and LeBA outperform both
counterparts (except for VGG-16). We will also provide their performance over defensive models in the revised paper.

*Writing Structure*: We will adjust the writing structure to make the paper self-contained, including simplifying the
related work, moving Algorithm A3 (LeBA) into main text, and experiment setting/ablation studies into appendix.

**R1:** We do appreciate your positive feedback and suggestions. On targeted attack, due to its intensive computational
cost, we provide the performance against Inception-V3 under 10,000 queries (same as the untargeted): SimBA (48.3%,
6465.4), SimBA++ (60.1%, 3472.6) and LeBA (66.7%, 4197.7). There is no surprise; both contributions introduced by
SimBA++ and LeBA still remain effective. We will report all results under 60,000 queries in the revised appendix.

**R2:** Thank you for insightful comments. We would like to emphasize that our method is intuitive yet effective; instead
of efficient gradient estimation with surrogate models (P-RGF), our method (SimBA++) significantly outperforms
previous SOTA, which inspires us to rethink the way to combine transferability-based and query-based attack. Besides,
we provide the first method to efficiently update the surrogate model with limited queries.

**R3:** Thank you for the very detailed suggestions for improving the structure and writing of our paper! We will carefully
revise our writing and language issues as per your advice. To respond your concerns, first, we would like to emphasize
that though the proposed SimBA++ is straightforward, it outperforms previous SOTA. This simple baseline could make
the community rethink the ways to use surrogate models. Second, the improvement of LeBA over SimBA++ seems
"marginal", but it is hard to obtain: SimBA++ has already outperformed previous SOTA by large margins, LeBA could
further improve the query efficiency consistently in all the experiments. We believe the setting of LeBA is reasonable in
practical scenario: we expect higher query attack efficiency with more query feedback obtained from victim models.
Please note that LeBA becomes effective just when query procedure begins (not after 1,000 queries totally completed).
As depicted in main Table 2, LeBA (*test*) is more efficient than LeBA (*training*). Third, we absolutely agree with you
that the threat model without surrogate models is also important, which is the reason why we emphasize the threat
model used in our paper (with surrogate models). Both threat models are theoretically and practically important .

As suggested, we report the performance of Square Attack Andriushchenko et al. (2020) on the same images as ours.
Please note that the attacked images are critical to the reported performance, therefore it is not suitable to compare them
directly. As shown in Table 1, our methods (both SimBA++ and LeBA) outperforms the Square Attack in most cases.
The brilliant ideas in Square Attack could also be integrated into ours (e.g., replace SimBA with Square Attack).

**R4:** Thank you for the golden comments! First, $L_2$ and $L_\infty$ threat models are both critical in practice. It is important to
select perturbation pixels for $L_2$ while not for $L_\infty$. Therefore, many black-box attack studies focus on only 1 threat
model (e.g., $L_2$ in SimBA). Even if both threat models are addressed in certain studies, the algorithms for $L_2$ and $L_\infty$
are different (e.g., P-RGF and Square Attack). We focus on $L_2$ threat model only in our study. Second, PGD (2000
steps) achieves 100%, 100%, 99.9% success for the defensive models in main Table 3, respectively. Note that JPEG
is not differentiable, we report the BPDA extension [2] of PGD. As suggested, it will be reported in main Table 3
as a reference for white-box attack. Third, we report the performance of NATTACK Li et al. (2019) in Table 1. As
NATTACK focus on the attack success over defensive models, we will also report its performance as per main Table 3.

# References

Andriushchenko, M. et al. Square attack: a query-efficient black-box adversarial attack via random search. In *ECCV*, 2020.

Li, Y. et al. Nattack: Learning the distributions of adversarial examples for an improved black-box attack on deep neural networks.
In *ICML*, 2019.


[Meta-Review · NeurIPS 2020]

This paper introduces a method to generate black-box adversarial examples by relying both on the transferability of adversarial examples, and a query-based mechanism. The reviewers were conflicted on the paper. All reviewers agreed on the strengths. The paper presents impressive results, and the attack is significantly stronger than the SimBA baseline from prior work. Reviewers also agreed on the differences, but diverged in their thoughts on if it was important. This paper is, in many ways, a collection of tricks that helps to improve the efficacy of SimBA. The authors dispute this claim in the response, but I agree with the reviewers. In both the abstract and the rebuttal, the authors argue that the scheme is simple and the techniques generalize. However, there is not any evidence that this is the case in the main body of the paper, and the supplied code does not justify this claim. Nevertheless, the strengths do outweigh the weaknesses. The strong results are important even if the paper does not (yet) show that the results will transfer. The strong results will be an important baseline for future work.